

# Screening of cytoprotectors against methotrexate-induced cytogenotoxicity from bioactive phytochemicals

Shaobin Gu[1,2], Ying Wu[1,3] and Jianbo Yang[4]

[1] College of Food and Bioengineering, Henan University of Science and Technology, Luoyang, China
[2] Luoyang Engineering and Technology Research Center of Microbial Fermentationon, Luoyang, China
[3] Henan Engineering Research Center of Food Material, Luoyang, China
[4] Rice Research Institute, Anhui Academy of Agricultural Science, Hefei, China

## ABSTRACT

As a well known anti-neoplastic drug, the cytogenotoxicity of methotrexate (MTX) has received more attention in recent years. To develop a new cytoprotector to reduce the risk of second cancers caused by methotrexate, an umu test combined with a micronucleus assay was employed to estimate the cytoprotective effects of ten kinds of bioactive phytochemicals and their combinations. The results showed that allicin, proanthocyanidins, polyphenols, eleutherosides and isoflavones had higher antimutagenic activities than other phytochemicals. At the highest dose tested, the MTX genetoxicity was suppressed by 34.03%~67.12%. Of all the bioactive phytochemical combinations, the combination of grape seed proanthocyanidins and eleutherosides from Siberian ginseng as well as green tea polyphenols and eleutherosides exhibited stronger antimutagenic effects; the inhibition rate of methotrexate-induced genotoxicity separately reached $74.7 \pm 6.5\%$ and $71.8 \pm 4.7\%$. Pretreatment of Kunming mice with phytochemical combinations revealed an obvious reduction in micronucleus and sperm abnormality rates following exposure to MTX ($p < 0.01$). Moreover, significant increases in thymus and spleen indices were observed in cytoprotector candidates in treated groups. The results indicated that bioactive phytochemicals combinations had the potential to be used as new cytoprotectors.

Corresponding author
Shaobin Gu, shaobingu@haust.edu.cn

## INTRODUCTION

Methotrexate (MTX) is one of the most intensively investigated and effective chemotherapeutic agents for inhibiting dihydrofolate reductase, resulting in depletion of tetrahydrofolate. However, the genotoxic effects of MTX have been reported in the somatic cells employing chromosome aberration and micronucleus test as the end points of evaluation (*Choudhury & Palo, 2004*). A present study has confirmed that MTX induced cytotoxic and genotoxic effects in the germ cells (*Padmanabhan et al., 2008*). Furthermore, MTX-treated RA patients have an increasing incidence of melanoma, non-Hodgkin's lymphoma, and lung cancer (*Buchbinder et al., 2008*). In addition, MTX chemotherapy for gestational trophoblastic tumors and acute lymphoblastic leukemia greatly increases the

risk of second tumors (*Schmiegelow et al., 2009*). Being a structural analogue of folic acid (FA), MTX competes with the normal substrate FA for the binding site on dihydrofolate reductase (DHFR), which is the critical enzyme involved in the synthesis of essential DNA precursors such as thymidylates and purines. Suppression of DHFR leads to depletion of tetrahydrofolates which are required for the synthesis of purines and thymidilate and thereby perturb the DNA synthesis (*Aggarwal et al., 2006*). On the other hand, the deficiency of folates induced by MTX interfering in the folate metabolism is often accompanied by genotoxic damage including strand breaks, chromosomal abnormalities, extensive incorporation of uracil in place of thymine into the DNA, defective DNA repair, anomalous DNA methylation patterns and increased somatic mutation rates (*Vinson & Hales, 2002*; *Branda et al., 2001*; *Knock et al., 2008*; *Kapiszewska et al., 2005*; *Branda et al., 2007*). In addition, Coleshowers's work indicated that MTX causes oxidative stress by reducing the activities of superoxide dismutase, catalase and glutathione reductase (*Coleshowers et al., 2010*). *Gressier et al. (1994)* demonstrated that MTX increases the amount of hydrogen peroxide and other free radicals which may lead to toxicity thus accelerating the rate of cellular damage. Several studies suggested that oxidative stress in the pathogenesis of MTX-induced damage play an important role in the various organs (*Vardi, Parlakpinar & Ates, 2012*). Thus, the development of efficient protective agents against methotrexate-induced cytogenotoxicity has attracted more and more attention.

In recent years, a number of natural plant products have been investigated as cytoprotectors to defend normal cells from the damage induced by MTX. *Horie et al. (2006)* reported that aged garlic extract could protect IEC-6 cells from the MTX-induced intestinal damage. *Verschaeve & Van Staden (2008)* found that apricot and $\beta$-carotene treatment could alleviate the impairment of oxidative stress and ameliorate MTX-induced intestine damage. *Chang et al. (2013)* showed that MTX-induced apoptosis of IEC-6 cells could be repressed by the pre-treatment of lutein. *Daggulli et al. (2014)* observed that carvacrol significantly reduced deleterious effects of MTX on testicular tissue. *Vardi, Parlakpinar & Ates (2012)* demonstrated that chlorogenic acid treatment may protect against the impairment of oxidative stress and ameliorate MTX-induced cerebellar damage in rats. It appears that proanthocyanidin from Vitis vinifera can protect the small intestine of rats and inhibits methotrexate-induced oxidative stress (*Gulguna et al., 2010*). There is evidence indicating that grape seed extract could ameliorate the MTX-induced oxidative injury in the rat liver (*Cetin et al., 2008*). Contrasting with garlic, apricot, *Origanum onites L.* panax ginseng, chlorogenic acid, proanthocyanidin and American ginseng, the cytoprotective effect of those bioactive phytochemicals against methotrexate-induced cytogenotoxicity has been rarely reported, such as eleutherosides (Siberian ginseng root), gingerols (ginger root), ginkgo flavone (ginkgo leaf), ginsenosides (ginseng root), polyphenols (green tea), polysaccharides (reishi mushroom), and isoflavones (soybean). Actually, ginger phenolic compounds exhibited potential antioxidant properties, moderate activity against xanthine oxidase, monoamine oxidase-A and $\alpha$-glucosidase, and protection of PC-12 and primary rat liver cells against $H_2O_2$-induced damage (*Peng et al., 2012*). Ginkgo flavonoids might protect against apoptosis of hippocampal neurons through inhibiting death receptor pathway or mitochondrial pathway underTNF-$\alpha$ background

(*Guo et al., 2015*). Ginsenoside can increase the activities of antioxidant enzymes such as T-SOD and GSH-Px and decrease the level of lipid peroxidation such as TBARS and protein carbonyl to block oxidative pathways (*Zhao, Li & Li, 2011*). Green tea polyphenols can protect against oxidative stress/damage and bladder cell death (*Coyle et al., 2008*). The antimutagenic and antioxidant properties of mushrooms polysaccharides have been also reported in literature (*Delmanto et al., 2001*; *Kozarski et al., 2011*; *Wang et al., 2015*). The antioxidant ability of soybean isoflavone is well known. However, most of the previous studies are focused on the cytoprotection of individual plant extracts or combined with other agents, such as Îš-carotene and quercetin. Studies on the protective potentials of various bioactive phytochemicals against MTX-induced damage, such as Siberian ginseng eleutherosides, chrysanthemum chlorogenic acid, ginger gingerols, grape seed proanthocyanidins, green tea polyphenols and so on, especially plant extract combinations have received less attention. To develop a more efficient protective agent against MTX-induced cytogenotoxicity, cytoprotective activity of ten different extracts and some extracts combinations were evaluated *in vivo* and *in vitro* tests. The combination of grape seed proanthocyanidins and eleutherosides from Siberian ginseng as well as green tea polyphenols and eleutherosides owned potentials to be used as new cytoprotectors.

## MATERIALS AND METHODS

### Materials

*S. typhimurium* TA1535/pSK1002 was kindly provided by Dr. Yoshimitsu Oda (Osaka Prefectural Institute of Public Health, Osaka, Japan). 4-NQO was used as a positive control. DMSO served as control and solvent. All bioactive phytochemicals (Chrysanthemum chlorogenic acid, garlic allicin, ginger root gingerols, ginkgo leaf flavone, ginseng root ginsenosides, grape seed proanthocyanidins, reishi mushroom polysaccharides, Siberian ginseng root eleutherosides, soybean isoflavones) were purchased from Changsha Active Ingredients Group Inc (Changsha, China). The purity of tested compounds was more than 95%. Kunming specific pathogen-free mice (4–6 weeks old, average body weight 19 ± 2 g) were provided by the Henan Laboratory Animal Center. License number: SCXK (Yu) 2005-0001. The present study was conducted in accordance with the principles outlined in the National Institutes of Health *Guide for the Care and Use of Laboratory Animals.*

### Umu test

To investigate protective effects of ten phytochemical compounds and their combinations against MTX induced Genotoxicity, an umu test was applied in the study. Doses of phytochemical compounds refer to the recommended dosage of Pharmacopoeia of the People's Republic of China (2010 English Edition). According to the compatibility theory of traditional Chinese medicine ( *Li & Xu, 2011*), pairwise combinations of phytochemical compounds with cytoprotective effects were prepared by dissolving two plant extracts in DMSO solvent. The active ingredient content of each phytochemical compound is 1 mg/ml in the plant extract combinations. The umu test was performed according to the method of Oda (*Oda et al., 1985*). The induction units (IU) of $\beta$-galactosidase activity, was calculated

according to the following equation:

$$IU = 1,000(A_{420} - 1.75 \times A_{550})/25 \times 0.1 \times A_{600}.$$

Induction ratio (R) was calculated according to the following equation:

$$R = \text{sample IU/control IU}.$$

## Assay of bone marrow micronucleus and indices of thymus, spleens in mice

The total of 32 mice were randomly divided into four groups with eight each group (four males and four females). The combination of bioactive phytochemicals was prepared by dissolving the two bioactive phytochemicals in DMSO, later diluted it with distilled water to an active ingredient content of 50 mg L$^{-1}$ for each bioactive phytochemical. The combination of bioactive phytochemicals was administered one week prior to the MTX exposure. Treatment group I: mice were given a combination of green tea polyphenols and eleutherosides from Siberian ginseng (0.2 ml/10 g W, i.g. once daily) for 15 days, and a single dose of MTX (2 mg kg$^{-1}$, i.p. once daily) was added on the 8th day. Treatment group II: mice were given a combination of grape seed proanthocyanidins and eleutherosides from Siberian ginseng for 15 days, and MTX was administered on the 8th day in a similar manner. Model group: animals received distilled water instead of bioactive phytochemicals combinations for 15 days and the same MTX protocol applied to this group on the 8th day. Control group: mice were given distilled water through 15 days and physiological saline instead of MTX was administered on the 8th day in a similar manner. Twelve hours after the final doses, the animals were euthanized by cervical dislocation.

The micronucleus assay was performed according to the method of *Schmid (1975)*. Femurs were removed from the animals, bone marrow extracted with fetal calf serum and maintained at 37 °C. The material was homogenized, transferred to a centrifuge tube and centrifuged at 1,000× g for 5 min. The supernatant was discarded and samples were prepared with the remaining cells. Then, samples were allowed to air dry and 24 h later they were fixed in absolute methanol for 5 min. They were then stained with 2% Giemsa stain diluted with distilled water. Slides were coded and from dry slides about 1,500 tinge blue coloured polychromatic erythrocytes were scanned from each animal, and the incidence of micronuclei in polychromatic (PCEs) and normochromatic (NCEs) erythrocytes was counted and the PCEs/NCEs ratios were scored.

The thymus and spleen indices were assayed according to the method (*Zhang et al., 2003*) and calculated as follows:

$$\text{thymus or spleen index} = \text{thymus or spleen weight/body weight} \times 1,000.$$

## Sperm deformity test in mice

Twenty-four male mice were randomly divided into four groups with six each group. MTX and the combination of bioactive phytochemicals were administered into the body by using the above-mentioned methods. Twelve hours after the final doses, the mice were

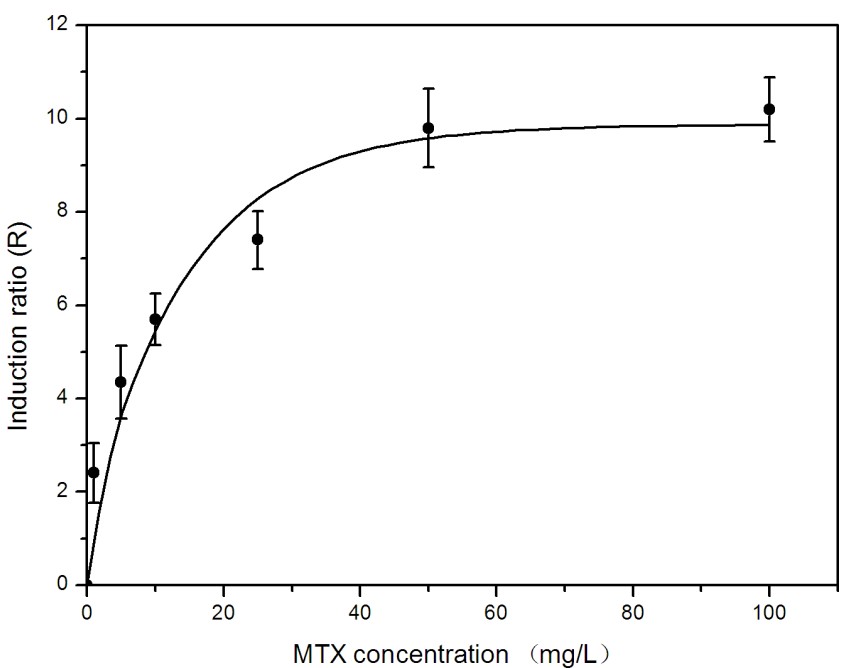

**Figure 1** Effects of methotrexate dose on umu gene expression in *S. typhimurium* TA1535/pSK1002.

sacrificed by cervical dislocation to get the bilateral epididymis. Sperm deformity test was carried out according to the following method of Wyrobek et al. Two sperm suspensions were prepared from the caudal of each testis by mincing the caudal in physiological saline (*Wyrobek et al., 1983*) . The sperm was spread on a slide glass and stained with 1% Eosine Y for 45 min after which the slides were air dried. A total of 1,000 sperm cells of mice were assessed for morphological abnormalities under oil immersion at ×1,000 magnification. Sperm head morphology was scored under the category of normal, sperm without hook, amorphous head, banana head and triangular head.

## Statistical analysis

Values are presented as means ± standard deviation (SD).The data were analyzed for statistical significance using *t*-test (SPSS 13.0 for Windows). A *p* value of less than 0.05 was deemed as significant.

## RESULTS

### Screening of cytoprotector derived bioactive phytochemicals based on SOS/umu test

To ensure the reliability of screening results of cell protective agent from bioactive phytochemicals plant extracts and avoid the interference on genetic toxicity test from the growth inhibition effect, it is necessary to select the appropriate concentration of MTX in SOS/umu test. Our previous study suggested that growth inhibition of *S. typhimurium* TA1535/pSK1002 was clearly observed while the concentration of MTX above 100 mg $L^{-1}$. Herein, we investigated the genotoxicity of MTX within the concentration range of 0–100 mg $L^{-1}$. Figure 1 showed that the dose–response relationship between umu

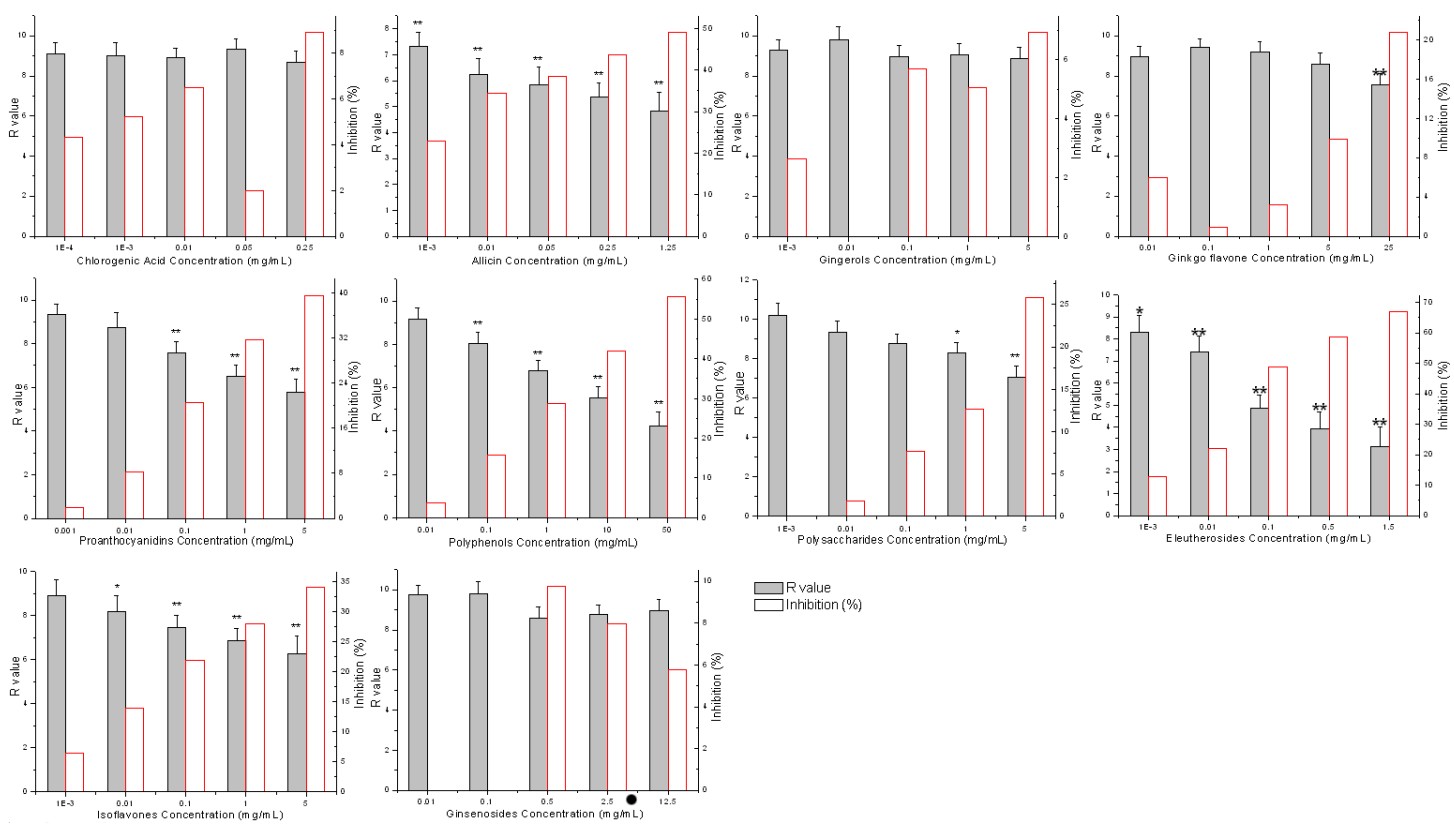

**Figure 2  Effects of bioactive phytochemicals on umu gene expression in *Salmonella typhimurium* TA1535/pSK1002 exposed to 50 mg L$^{-1}$ methotrexate.** Inhibition (%) $= 100 \times (R_{\text{control}} - R_{\text{sample}})/R_{\text{control}}$; $^*p < 0.05$, $^{**}p < 0.01$.

gene expression level and MTX genetoxicity wasn't linear relationship. Before the MTX concentration approached 50 mg L$^{-1}$, *R* value has already reached the platform. This result is similar to the kinetics of induction of the umu operon by mitomycin C in *S. typhimurium* NM2009 (*Oda et al., 1995*). In order to ensure the reliability of screening results of cell protective agent from plant extracts, 50 mg L$^{-1}$ MTX was chosen in subsequent SOS/umu test.

Effects of ten different plant extracts on umu gene expression in *Salmonella typhimurium* TA1535/pSK1002 exposed to MTX were shown in Fig. 2. It was demonstrated that allicin, proanthocyanidins, polyphenols, eleutherosides and isoflavones showed stronger antimutagenic activities than other five kinds of plant extracts. At the highest dose tested, the MTX genetoxicity was inhibited by 34.03%~67.12%. Subsequently, the pairwise combinations of the five bioactive phytochemicals with antimutagenic activities were prepared by dissolving two plant extracts in DMSO solvent. The concentration of effective components of each extracts reached 1 g L$^{-1}$. According to the above methods, antimutagenic activity of various combinations of plant extracts was determined by umu test. Antimutagenic potential of plant extract combinations were illustrated in Fig. 3. Of all the plant extract combinations, the combination of grape seed proanthocyanidins and eleutherosides from Siberian ginseng as well as green tea polyphenols and Siberian ginseng

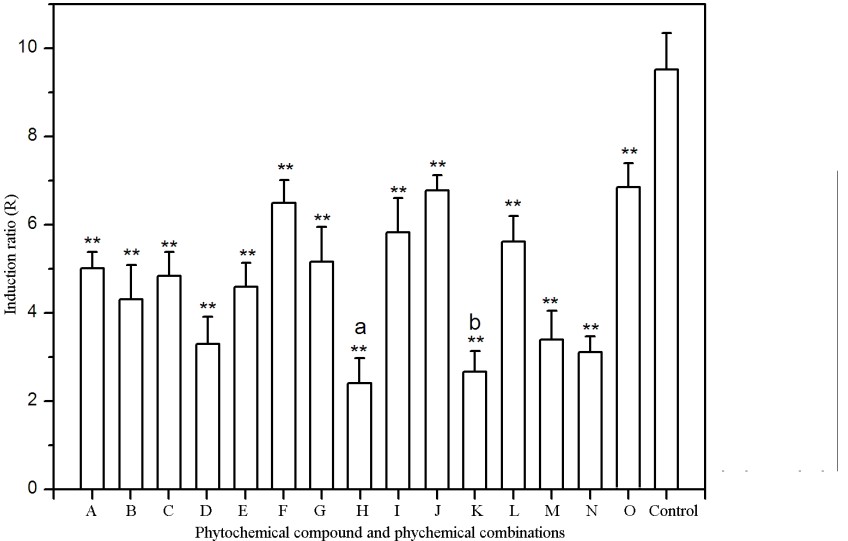

**Figure 3** **Effects of phytochemicals and phytochemical combinations on umu gene expression in S. typhimurium TA1535/pSK1002 exposed to 50 mg L$^{-1}$ methotrexate.** A, Allicin (1 g L$^{-1}$); B, Allicin + Grape seed proanthocyanidins; C, Allicin + Green tea polyphenols; D, Allicin + Eleutherosides; E, Allicin + Soybean isoflavones; F, Grape seed proanthocyanidins (1 g L$^{-1}$); G, Grape seed proanthocyanidins + Green tea polyphenols; H, Grape seed proanthocyanidins + Eleutherosides; I, Grape seed proanthocyanidins + Soybean isoflavones; J, Green tea polyphenols (1 g L$^{-1}$); K, Green tea polyphenols + Eleutherosides; L, Green tea polyphenols + Soybean isoflavones; M, Eleutherosides (1 g L$^{-1}$); N, Eleutherosides + Soybean isoflavones; O, Soybean isoflavones (1 g L$^{-1}$). a, $p < 0.01$ as compared to Eleutherosides treatment group; b, $p < 0.05$ as compared to Eleutherosides treatment group; *$p < 0.05$, **$p < 0.01$ as compared to control.

extract exhibited higher cytoprotective activity. The inhibition rate of methotrexate-induced genotoxicity separately reached $74.7 \pm 6.5\%$ and $71.8 \pm 4.7\%$. The both combinations of plant extracts were selected as cell protective agent candidates, and the cytoprotective effects of candidates would be subsequently assessed by micronucleus test and sperm malformation test *in vitro*.

## Evaluation of the antimutagenic potentials of cytoprotector candidates by micronucleus test

The micronucleus assay is internationally recognized as the standard method to detect the mutagenicity of chemicals. In order to assess the protective effects of the candidate cytoprotectors against the genotoxicity caused by MTX, micronucleus assay was employed in the next trial. With administration of the candidate plant extract combinations, the data of the bone marrow polychromatic erythrocyte micronucleus test in mice exposed to MTX were presented in Fig. 4. Whether male or female, there were statistically significant increases ($p < 0.01$) in the frequency of micronucleated polychromatic erythrocytes and ratio of polychromatic erythrocytes (PCE) to normochromatic erythrocytes (NCE) in model group and control group. However, the treatment of the candidate cytoprotectors markedly restrained the incidence of mice bone marrow micronucleus, and improved the ratio of PEC and NEC. Furthermore, there were significant differences between treatment groups and model group ($p < 0.01$). In terms of the inhibition of micronucleus formation, the

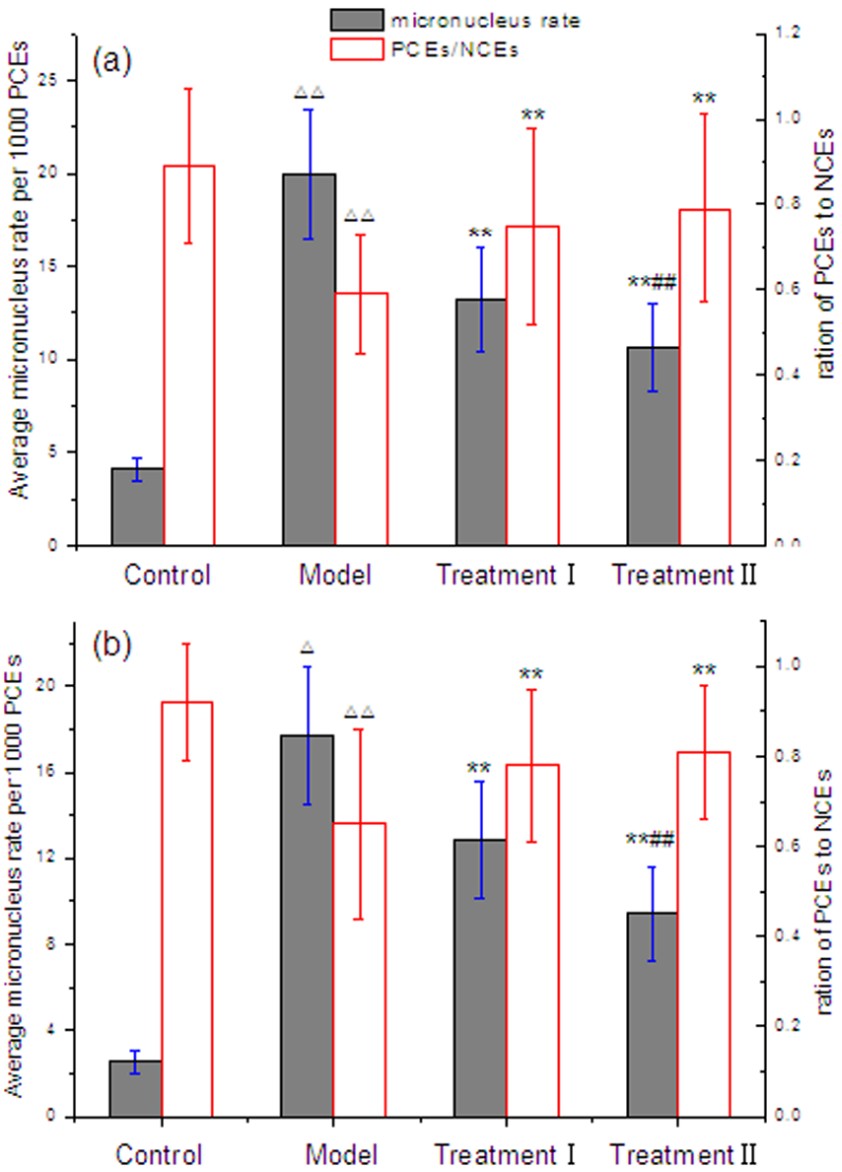

**Figure 4** **Effect of cytoprotector candidates on incidence of micronucleated polychromatic erythrocytes in bone marrow cells of mice treated with methotrexate.** (A) male mice; (B) female mice; (1) Control Group = saline; Model Group = methotrexate + normal saline; Treatment Group I = methotrexate + combination of polyphenols and eleutherosides; Treatment Group II = methotrexate + combination of proanthocyanidins and eleutherosides. (2) $\Delta P < 0.05$, $\Delta\Delta P < 0.01$ vs control. (3) *$P < 0.05$, **$P < 0.01$ vs Model Group. (4) #$P < 0.05$, ##$P < 0.01$ vs Treatment I.

combination of grape seed proanthocyanidins and eleutherosides from Siberian ginseng was superior to the combination of green tea polyphenols and eleutherosides. Meanwhile, the two treatment groups did not cause differences between male and female mice. It could be considered that the two combinations had promising effects on suppression of MTX-induced micronuclei in mice bone marrow cells.

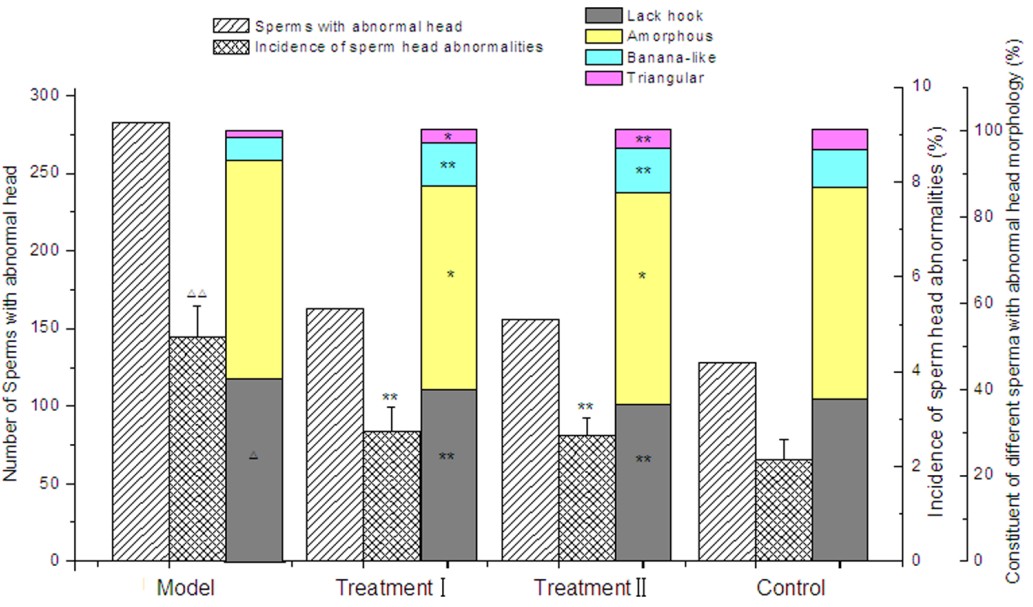

**Figure 5** **Effect of cytoprotector candidates on abnormalities of male Kunmin mice sperm head after consecutive 7 days of methotrexate exposure.** (1) Control Group = saline; Model Group = methotrexate + normal saline; Treatment Group I = methotrexate + combination of polyphenols and eleutherosides; Treatment Group II = methotrexate + combination of proanthocyanidins and eleutherosides. (2) $\Delta P <$ 0.05, $\Delta\Delta P < 0.01$ vs control. (3) $*P < 0.05$, $**P < 0.01$ vs Model Group.

## Influence of cytoprotector candidates on reproductive toxicity induced by MTX

To investigate whether the combinations of bioactive phytochemicals treatment could strengthen or weaken the reproductive toxicity induced by MTX, sperm tests had been adopted. The method is one reliable and easy way to detect reproductive toxicity. In this study, the kinds of abnormal sperm in any group were mainly no hooks, amorphous, bananas and triangular head heads. As could be seen from Fig. 5, the incidence of mouse sperm head deformity of treatment groups had no significant difference from that of the control ($p > 0.05$). Significant differences could be observed between the treatment groups and model groups ($p < 0.01$). The summary of no hooks and amorphism heads in the model accounted up to 90% of the total sperm head morphology, significantly higher than that of control group and treatment group. The results illustrated that the two combinations of bioactive phytochemicals may alleviate the reproductive toxicity caused by MTX. Compared with the model group, the percentage of different types of abnormal sperm showed significant difference. The percent of Lack hook, Amorphous, Banana-like and triangular in Treatment I were 39.9%, 47.2%, 9.8% and 3.1%, respectively. In the case of sperm aberration, the result of the Treatment II was closes to the data of control group. It seemed that cytoprotective activities of proanthocyanidins and eleutherosides combination were superior to that of polyphenols and eleutherosides.

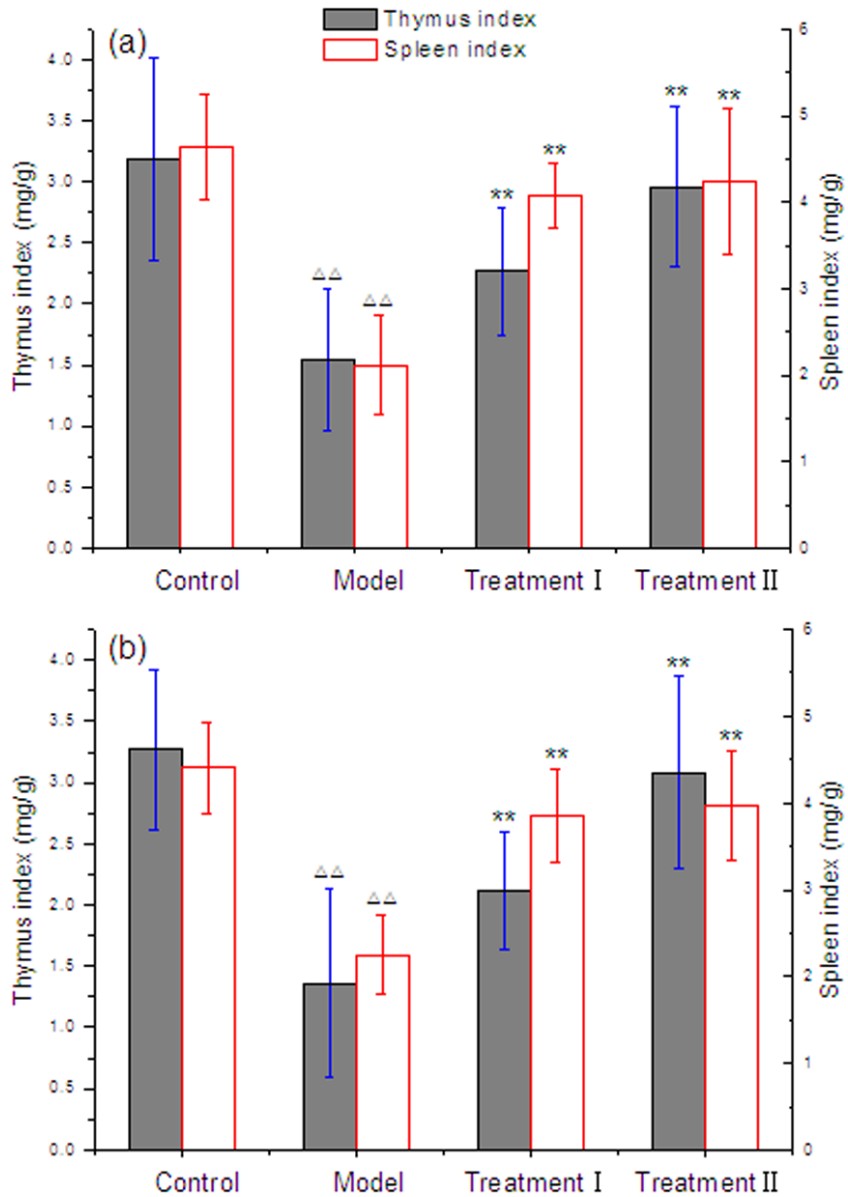

**Figure 6** Effect of cytoprotector candidates on thymus and spleen indices of mice exposed to Methotrexate. (A) male mice; (B) female mice; (1) Control Group = saline; Model Group = methotrexate + normal saline; Treatment Group I = methotrexate + combination of polyphenols and eleutherosides; Treatment Group II = methotrexate + combination of proanthocyanidins and eleutherosides. (2) $\Delta P < 0.05$, $\Delta\Delta P < 0.01$ vs control. (3) *$P < 0.05$, **$P < 0.01$ vs Model Group.

## Influence of cytoprotector candidates on immune organ indices

As cytoprotector candidates, the effect of two combinations of bioactive phytochemicals on thymus and spleen indices of mice exposed to MTX was shown in Fig. 6. The thymus and spleen indices were similar between control and the treatment groups in both male and female mice. MTX exposure reduced thymus and spleen indices of mice. *Kawai (2014)* reported that MTX could markedly decrease white blood cells, thymic and splenic

lymphocytes at dose ≥5 mg/kg. However, there was a significant difference between the treatment plus control group and the model group ($p < 0.01$). The combination of grape seed proanthocyanidins and Siberian ginseng eleutherosides obviously diminished the effects of MTX exposure on indices of thymus and spleens in mice. And, the increase of spleens index in the treatment of grape seed proanthocyanidins and Siberian ginseng eleutherosides combination was higher than that of green tea polyphenols and Siberian ginseng eleutherosides combination administration. Our findings implied that the two combinations of bioactive phytochemicals not only could affect the immune organ indices of experimental animal Kunming mice, but also could effectively relieve the immune toxicity caused by MTX.

## DISCUSSION

The effective estimating for antimutagenic properties of bioactive phytochemicals includes the bacterial gene mutation assay Ames test (*Verschaeve & Van Staden, 2008*; *Horn & Vargas, 2003*), mammalian cell culture benzo(a)pyrene metabolism assay (*Cassady et al., 1988*), the mammalian cell gene mutation assay (*Mersch-Sundermann et al., 2004*), and the *in vitro* micronucleus assay (*Serpeloni et al., 2008*). Compared with animal-cell-based systems, microbe-based assays for screening cytoprotectors present several advantages, such as the simplicity of the procedures, the relatively short time needed to obtain results, and cost-effective. Nevertheless, except for the Ames test, there are few reports in the literature describing the studies of evaluation for antimutagenic properties of bioactive phytochemicals by umu test so far. In this study, we selected a short-term bacterial test system, the umu test, to evaluate the antimutagenic potential of plant extracts. Moreover, based on comparison of the umu test results (486 chemicals) with the Ames test (274 compounds) as well as rodent carcinogenicity data (179 compounds), *Reifferscheid & Heil (1996)* found good agreement between the umu test and the Ames test results. Thus umu test could be developed an effective high-throughput screening assay to evaluate antimutagenic potential of phytochemicals same as Ames test and comet assay.

Natural plant medicine is an important resource to develop new cytoprotectors. Cytoprotective effects of many bioactive phytochemicals have been proved in the past several decades. Isoflavones from soybean seeds have showed antimutagenic activity in *S. typhimurium* TA1535/pSK1002 and TA100 (*Miyazawa et al., 1999*). Garlic extract has been proved that it could reduce apoptotic cell injury induced by MTX. It should be noted that garlic was most popular supplement in US households (*Amagase et al., 2001*). Meanwhile, polyphenols from green tea (Chinese Gunpowder and Japanese Sencha) exhibited high antimutagenic activity in the Ames test as well as in *S. cerevisiae* D7 test. Moreover, in the peripheral blood lymphocytes method reduced number of abberant cells as well as decreased number of chromosome breaks was observed using both green tea extracts (*Bunkova, Marova & Nemec, 2005*). What's more, grape seed extract played a role in attenuating the genotoxicity induced by cisplatin (*Attia et al., 2008*). Contrasting with allicin, proanthocyanidins, polyphenols and isoflavones, chlorogenic acid, proanthocyanidin and American ginseng, cytoprotective effect of those

bioactive phytochemicals against methotrexate-induced cytogenotoxicity has been reported rarely, such as Siberian ginseng eleutherosides, ginger gingerols, ginkgo flavone, ginseng ginsenosides, mushroom polysaccharides, and soybeam isoflavones. Our results suggested that allicin, proanthocyanidins, polyphenols, eleutherosides and isoflavones showed stronger antimutagenic activities than other five kinds of plant extracts. Moreover, it was demonstrated results single eleutherosides and its combinations with proanthocyanidins, polyphenols had strong antimutagenic effect. It is well known that the detrimental effects of MTX was partly due to its direct toxic action by increasing reactive oxygen species production, although the exact mechanisms of methotrexate-induced toxicity had not yet been elucidated to date (*Oktem, 2006*). Coleshowers's work indicated that MTX causes oxidative stress by reducing the activities of superoxide dismutase, catalase and glutathione reductase (*Coleshowers et al., 2010*). *Gressier et al. (1994)* demonstrated that MTX increases the amount of hydrogen peroxide and other free radicals which may lead to toxicity thus accelerating the rate of cellular damage. Several studies have well confirmed the contribution of oxidative stress in the pathogenesis of MTX-induced damage in the various organs (*Vardi, Parlakpinar & Ates, 2012*). It is supposed that the oxidative stress induced by MTX was reduced and the cell damage caused by free radical accumulation was alleviated owing to potential antioxidant activity of the grape seed proanthocyanidins, green tea polyphenols, soybean isoflavone and Siberian ginseng eleutherosides. *Gulguna et al. (2010)* demonstrated that proanthocyanidin from *Vitis vinifera* can inhibits methotrexate-induced oxidative stress. *Yu et al. (2016)* found that a polyphenol-rich Herb (*Scutellariae radix*) ingestion increased the systemic exposure and mean residence time of MTX via modulation on efflux transporters multidrug resistance–associated protein 2 and breast cancer resistance protein. *Chiang et al. (2005)* observed that the coadministration of Pueraria lobata root decoction (an isoflavone-rich herb) significantly decreased the elimination and resulted in markedly increased exposure of MTX in rats. Recently, it was proved that eleutherosides (as one kind of polyphenols) were promising chemical substances with antioxidant properties. Substantially, grape seed proanthocyanidins, green tea polyphenols, soybean isoflavone and Siberian ginseng eleutherosides belong to the members of polyphenols which have strong antioxidant capacity. In this radical scavenging mechanism, polyphenols sacrificially reduce ROS/RNS, such as $^{\bullet}OH$, $O_2^{\bullet-}$, $NO^{\bullet}$, or $OONO^-$ after generation, preventing damage to biomolecules or formation of more reactive ROS (*Perron & Brumaghim, 2009*). Previous research found that water extract of Siberian ginseng showed significant antioxidant activity and protective effect against oxidative DNA damage induced by $H_2O_2$ (*Park et al., 2006*). Moreover, the methanol extract of root, stem, and leaf of *Eleutherococcous senticosus* showed inhibitory effects on the mutagenicity induced by 2-AF or Trp-P-1 in *S. typhimurium* TA98 (*Park et al., 2002*). Therefore, we inferred that the cytoprotector candidates may play a key role in attenuating the methotrexate-induced cytogenotoxicity due to their antimutagenic and antioxidant activity.

In addition, the previous study had proved that MTX could bring out the reproductive toxicity (*Padmanabhan et al., 2008*). MTX treatment significantly reduced the sperm count and increased the occurrence of sperm head abnormalities. However, the administration of phytochemicals combinations to Kunming specific pathogen-free mice appeared clearly

decreases in sperm abnormality rate in the case of MTX exposure. *Akram et al. (2012)* ever reported that American ginseng extract treatment exhibited therapeutic effects on sperm parameters in rats treated with Cyclophosphamide (CP), which is an antineoplastic agent and immunosuppressive medicine in the treatment of various types of tumors, and autoimmune diseases such as systemic lupus erythematosus, rheumatoid arthritis and multiple sclerosis (*Tripathi & Jena, 2009*). Moreover, ginseng has been demonstrated to have a cytoprotective effects against these toxins, in which administration of *Panax ginseng* extract was reported to markedly decrease the 2,3,7,8-tetrachlorodibenzo-p-dioxin-induced pathological and genotoxical damages in rat testes (*Lee et al., 2007*). Recent studies had confirmed that green tea and soybean extracts showed protective effects against reproductive toxicity induced by CP. *Fahmy et al. (2014)* found a significant decrease in the percentage of sperm abnormalities in orally administrated soybean extracts. *Zanchi et al. (2015)* demonstrated that polyphenols improved CP-induced damage on reproductive system, and its effect is probably due to high concentrations of catechins and antioxidant activity.

Although some evidence suggested that Siberian ginseng may not stimulate immune function (*Wang et al., 2003*), it has been indicated that polyphenols from green tea and grape seed proanthocyanidin have apparent immunomodulatory effects. Sheikhzadeh found that decaffeinated green tea in lower doses of administration could enhance the immunity of rainbow trout (*Sheikhzadeh et al., 2011*). The dietary administration of green tea supplementation positively raises the cellular immune responses and disease resistance of kelp grouper Epinephelus bruneus to Vibrio carchariae (*Harikrishnan, Balasundaram & Heo, 2011*). *Tong et al. (2011)* demonstrated that grape seed proanthocyanidins (GSPs) could improve functional activation of the immune system, and the antitumor effects of GSPs were achieved by immunostimulating properties. Our findings implied that the two combinations of phytochemical compounds not only could affect the immune organ indices of experimental animal Kunming mice, but also could effectively relieve the immune toxicity caused by MTX. According to the above results, it can be deduced that the combination of grape seed proanthocyanidins and Siberian ginseng eleutherosides as well as green tea polyphenols and eleutherosides was able to weaken the reproductive toxicity induced by MTX.

## CONCLUSIONS

In this study, we developed an effective screening assay to evaluate the antimutagenic potential of bioactive phytochemicals based on an umu assay coupled with a micronucleus test. Allicin, proanthocyanidins, polyphenols, eleutherosides and isoflavones showed higher antimutagenic activity than other phytochemicals. Moreover, of all the bioactive phytochemicals combinations, the combination of proanthocyanidins and eleutherosides as well as polyphenols and eleutherosides obtained higher cytoprotective effects. In the case of MTX exposure, the administration of the two combinations of bioactive phytochemicals to Kunming mice clearly presented decreases in micronucleus induction and in sperm abnormality rate. Moreover, obvious increases in thymus and spleen indices were observed

in the cytoprotector candidates treated groups. The results implied that phytochemicals combination of proanthocyanidins, eleutherosides and polyphenols could be used as new cytoprotectors.

## ACKNOWLEDGEMENTS

We are grateful to Dr. Yoshimitsu Oda for the generous gift of *S. typhimurium* TA1535/pSK1002.

### Funding

This work was supported by the Foundation for University Key Teacher by Henan Province (Grant No. 2014GGJS-056) and the National Natural Science Foundation of China (Grant No. U1304307). The funders had no role in study design, data collection and analysis, decision to publish, or preparation of the manuscript.

### Grant Disclosures

The following grant information was disclosed by the authors:
Henan Province: 2014GGJS-056.
National Natural Science Foundation of China: U1304307.

### Competing Interests

The authors declare there are no competing interests.

### Author Contributions

- Shaobin Gu conceived and designed the experiments, performed the experiments, analyzed the data, contributed reagents/materials/analysis tools, wrote the paper, prepared figures and/or tables, reviewed drafts of the paper.
- Ying Wu performed the experiments, contributed reagents/materials/analysis tools, reviewed drafts of the paper.
- Jianbo Yang conceived and designed the experiments.

### Animal Ethics

The following information was supplied relating to ethical approvals (i.e., approving body and any reference numbers):

Kunming specific pathogen-free mice were provided by the Henan Laboratory Animal Center. License number: SCXK (Yu) 2005-0001.The present study was conducted in accordance with the principles outlined in the National Institutes of Health Guide for the Care and Use of Laboratory Animals.

### Data Availability

The raw data has been supplied as Data S1.

## Supplemental Information

Supplemental information for this article can be found online at http://dx.doi.org/10.7717/peerj.1983#supplemental-information.

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
