# Peer review of "Screening of cytoprotectors against methotrexate-induced cytogenotoxicity from bioactive phytochemicals"

_PeerJ, doi:10.7717/peerj.1983_

## Round 0.1 · original submission · Major Revisions

This manuscript has been reviewed by three independent reviewers. While the work and the key findings have good potential, this manuscript is not acceptable in this format. The reviewers have raised questions on the methodology, additional experiments and major rewriting.

Specifically, the main objective of this study needs to be focused on the context of molecular mechanism, if not for all compounds, but for some key combinations. If you need more time, please let us know.

Reviewer 1 ·

Basic reporting

In abstract phytochemicals combinations will be phytochemical combinations.
In 119 line in vitro will be italicized i.e. in vitro
In Figure 1 S. typhimurium will be italicized i.e. S. typhimurium
In figure X Axis Caption is absent, only, y axis caption is present.

Experimental design

Is acceptable

Validity of the findings

Resonable

Additional comments

Please See and change typological errors as mentioned above as examples.
Please See and check the references.

·

Basic reporting

No comments

Experimental design

Need clarification

Validity of the findings

good

Additional comments

The manuscript entitled “Screening of cytoprotector against methotrexate (MTX) - induced cytogenotoxicity from bioactive phytochemicals by umu test combined with micronucleus Assay” by Gu et al. reports the evaluation of the cytoprotective effect of ten different kinds of plant derived phytochemicals and their combinations and discuss their possible use as cytoprotectors to reduce the risk of secondary cancers caused by MTX-induced cytogenotoxicity. They demonstrated in Kunming mice that some of phytochemicals (namely allicin, proanthocyanidins, polyphenols, eleutherosides, and isoflavones) have higher antimutagenic activities as inhibitor of MTX genotoxicity. Interestingly, they found stronger antimutagenic effects when different phytochemicals used in combination (combination of grape seed proanthocyanidins with eleutherosides from siberian ginseng as well as green tea polyphenols with eleutherosides). Though the work is interesting, authors need to address and clarify the following concerns -
1. How the phytochemicals were extracted and from which part? In which form those have been used? How the purity levels of the phytochemicals have been ensured. Impurities might be making in non-reproducible with additional problems.
2. What was the scientific basis of selection of phytochemicals and their combinations? Details should be given in materials methods and mentioned in relevant places of results.
3. In table 3 only treatment types should be specified. Different types of sperm abnormalities (which rather show some inconsistency in model and treatment) are not discussed.
4. How were the doses of the phytochemicals established? How was their effect on different cell type (without MTX)? Many phytochemicals also have side effects on animal if applied in excess doses. How was it ruled out?
5. What was the basis of p value < 0.01 as a cut-off value?
6. Figure 1 X and Y axes should be clearly specified.
7. It requires improvement of writing, materials method, results and discussion part.

Reviewer 3 ·

Basic reporting

The manuscript entitled “Screening of cytoprotector against methotrexate-induced cytogenotoxicity from bioactive phytochemicals by 3 umu test combined with micronucleus Assay” is a well-designed manuscript and tables are self-explanatory. However, in my opinion, there should be few more experimental data that need to be added before it can be accepted finally. I recommend this MS for major revision.
This paper describes the importance of different phyto-components (even belong to different groups) in the context of cytoprotection against a specific chemotherapeutic agents namely against Methotrexate. The authors have used different model system to establish the positive effects of these compounds.
The importance of these phyto components are well known, though this MS re-establish the importance of these compounds, positively through a more systematic analysis. However, the molecular mechanism by which these compounds exert cytoprotection in different model systems has not been investigated/described. This portion need attention. A major drawback of this MS is the lack of in-depth analysis into the mechanisms. The authors should take at least one of the many promising compounds and shade the molecular mechanism involving that. As these compounds belong to different groups, it is difficult to assume a single pathway/mechanism is involved in all the model systems. Never-the-less, the manuscript should touch these aspects.

Experimental design

There are certain additional experiments needed to improve this MS.

a) The phyto compounds were applied few days before applying Methotrexate. What happens if these compounds were added at the same time and/or after application of Methotrexate? Does these compounds exert the same cytoprotection?
b) The time gap between Methotrexate application at the 8th day and collection of cells/samples for analysis is only 12 hours. Why such a short time period? Is it due to the fact that beyond 12 hours these compounds are not effective? That need to be addressed. What will happen if the time gap is increased? In fact, if aimed to improve the “human application/human life”, then it is expected to have a much prolonged effect. Because in case of human application, once the Methotrexate is applied, the cells/tissue will be exposed to the Methotrexate for a long time (in contrast to 12 hours). Therefore, the authors must perform a prolonged study to check how effective is these compounds against Methotrexate exposure in terms of time gap.
c) The authors have used only Methotrexate as an agent. If these phyto compounds are really offering “cytoprotection” than need at least another chemotherapeutic agent such as Taxol. If the authors choose some of the key combinations against another chemotherapeutic agent, that will be more convincing.

Validity of the findings

As mentioned before.

Additional comments

Besides, taking care of the above mentioned points, the authors needs to add why these phytochemicals were chosen for the study (inclusion criteria). Also there are descriptions about how these compounds can act on the biological systems (such as specific protein (receptors/channels/enzymes etc., ROS signalling, etc). The authors need to comment on these.

---

## Round 0.2 · Minor Revisions

Dear Authors,

Your manuscript has been re-reviewed by two of the original reviewers and both think that the revised version is much better than the previous version. However, they pointed out that this manuscript needs attention in terms of typos and English language as a whole.

Also if you can clarify the comments from reviewer 3, regarding the "rebuttal letter", that will be helpful.

Reviewer 1 ·

Basic reporting

No comments

Experimental design

Experimental design is fine.

Validity of the findings

The findings are valid and in fact potential to try many different combinations of plant extracts and in different concentrations.

Additional comments

There are still many errors in English and typos. such as
line number 117 - Thirty - two
line number 261 - Vitis vinifera should have been italics etc.

The author should closely look at the typos.

Reviewer 3 ·

Basic reporting

The revised version is certainly better and much improved than the previous version. In the revised version the authors have addressed most but not for all the comments by performing. In the rebuttal letter, the authors have mentioned comments such as “A greater improvement manuscript will be submitted”, “According to the reviewer’s advice, we will perform a prolonged study to check how effective is these compounds against Methotrexate exposure in terms of time gap in the following experiments”, “A greater improvement manuscript will be submitted”. Such comments are bit confusing and may indicate that the authors are probably trying to submit another manuscript with more details. I do understand that there are certain area which probably needs a longer time frame and therefore may be beyond the scope of this manuscript.



In addition, this manuscript certainly needs language editing and scientific writing editing. For example, some of the scientific names appear in the pdf file (caption of Figure 1 and Figure 2) are not written in italics.

Experimental design

OK

Validity of the findings

OK

---

## Round 0.3 · accepted · Accept

Dear Authors,

I am happy to inform that your manuscript has been accepted for publication in PeerJ. However, at this stage I insist that you take help of our professional staff members to ensure that your manuscript is free from any errors such as language, spellings and grammatical mistakes.